# Does a High-Risk (>1/50) Result for First-Trimester Combined Screening Always Entail Invasive Testing? Which Patients from This Group Might Benefit from cfDNA Testing?

**DOI:** 10.3390/biomedicines10102579

**Published:** 2022-10-14

**Authors:** Rocío García-Jiménez, Irene Valero, Isabel Corrales-Gutiérrez, Reyes Granell, Carlota Borrero, José Antonio Sainz-Bueno

**Affiliations:** 1Obstetrics and Gynecology Department, Juan Ramon Jiménez Hospital, 21005 Huelva, Spain; 2Obstetrics and Gynecology Department, Virgen Macarena Hospital University, 41009 Seville, Spain; 3Obstetrics and Gynecology Department, Faculty of Medicine, University of Seville, 41009 Seville, Spain; 4Obstetrics and Gynecology Department, Valme University Hospital, 41014 Seville, Spain

**Keywords:** cell-free DNA, first-trimester combined testing, contingent screening, aneuploidy, non-invasive prenatal testing, trisomy 21, trisomy 18, trisomy 13

## Abstract

Currently, cell-free DNA (cfDNA) is offered as part of a contingent screening for patients with a first-trimester combined test (FCT) risk between 1/50 and 1/250. However, most aneuploidies are within the group of patients with a risk above 1/10. An observational, retrospective, and multi-centric study was carried out, to evaluate the theorical performance of lowering the cut-off point for the high-risk group from 1/50 to 1/10. Out of the 25,920 patients included, 25,374 (97.9%) consented to the cfDNA contingent screening for aneuploidies. With the proposed strategy, knowing that the detection rate (DR) of cfDNA testing for trisomy 21 is 99.7%, the DR for trisomy 21 would have stayed in a 93.2%, just as it was with the current strategy. In this instance, 267 (1.1%) invasive tests would have been performed, while the current strategy had a total of 307 (1.2%). The false positive rate (FPR) rate would have stayed at 5.2% in both scenarios. In conclusion, the contingent screening of aneuploidies based in the result of the FCT, offering the analysis of cfDNA to patients with an intermediate risk after lowering the cut-off point from 1/50 to 1/10, is a valid alternative that might maintain the current detection rates and avoid the complications associated with invasive testing.

## 1. Introduction

Prenatal detection of genetic anomalies continues to be a challenge for experts in Fetal Medicine. The definitive diagnosis is made with invasive testing (IT), such as chorionic villus sampling or amniocentesis, whose success depends on the experience of the operator and the clinic. These procedures entail the risk of complications, with fetal loss being the most crucial one. Although fetal loss rates usually range between 0.5–1%, recent reports estimate it to be between 0.1–0.2% for both techniques, while the remaining losses are related to the reason that caused the IT to be performed in the first place [1,2]. The economic costs of these procedures, along with the risk of complications [1,2,3,4], have prevented its implementation as universal screening [5]. Hence, several screening strategies have been proposed with non-invasive testing (NIPT) in order to maintain an adequate detection rate (DR) without needing to perform more IT and maintaining the same costs.

Up until recently, the first-trimester combined test (FCT) was the first-line strategy for the screening of trisomies 21, 18 and 13 in Spain. The FCT uses a combination of maternal age, fetal nuchal translucency (NT) thickness, serum-free β-human chorionic gonadotropin (β-hCG) and pregnancy-associated plasma protein A (PAPP-A). The DR of the FCT for trisomy 21 is 90%, while it is around 95% for both trisomies 18 and 13, with an overall false positive rate (FPR) of 5% [6].

The recent emergence of cell-free DNA (cfDNA) testing in maternal blood for screening of fetal trisomies caused a paradigm shift, given its reported DR of 99.7%, 97.9% and 99% for trisomies 21, 18 and 13, respectively, as early as 10 weeks’ gestation, with an FPR of 0.12% [7,8]. Despite this being a simple technique, which only requires a maternal blood sample, and its highly effective detection of trisomies, it has not been implemented as universal screening test due to its higher cost in comparison with the classic FCT [9,10].

Some groups have attempted to establish a contingent screening, with cfDNA testing performed in an intermediate-risk group, in order to reduce the IT rates while maintaining the DR. Recent publications have proven that this is a valid approach, using several cut-off points [11,12,13]. The current guidelines of the Spanish Society of Gynecology and Obstetrics (SEGO) has incorporated cfDNA testing as part of a contingent screening based on the results of the FCT. Thus, it is recommended to perform universal screening with the FCT, followed by a contingent cfDNA test for those with an intermediate risk between 1/50 and 1/250, at the time of the blood sampling, or 1/270 at the time of the delivery, and only those with a high-risk FCT above 1/50 are offered an IT [5]. This strategy allows us to maintain the DR [11,14] without increasing the economic costs [15]. However, it has been described that most aneuploidies are within the group of patients with a risk above 1/10 or with an increased thickness of the NT [13]. Thus, patients with a risk between 1/10 and 1/50 are exposed to the complications of IT, although its risk for trisomies is significantly lower. Knowing the high DR of cfDNA for trisomies, we wondered if these patients might benefit from cfDNA testing in the same way as the intermediate-risk group. Thus, our main objective was to evaluate the plausibility and theorical performance of lowering the cut-off point for the high-risk group from 1/50 to 1/10 within the current contingent screening strategy, while maintaining the DR and FPR.

## 2. Materials and Methods

An observational, retrospective, and multi-centric study was carried out, including the data of 25,920 patients with singleton pregnancies who attended their first-trimester hospital visit at one of the three participant hospitals in southern Spain. The data collected were included during the timeframe from the implementation (January 2018) of the contingent first-trimester screening with cfDNA, in each center, until July 2020. The study received the approval of the local bioethics committee (0109-N-16). The inclusion criteria were patients with singleton pregnancies who attended their first-trimester hospital visit at 11–13 weeks’ gestation. Patients with a multiple gestation and patients who initiated follow-up at 14 weeks or later were excluded from the study. 

### 2.1. Implementation of cfDNA Contingent Screening for Aneuploidies

The three participant hospitals apply the cfDNA contingent screening for aneuploidies, performing a first-trimester ultrasonography evaluation at 11–13 weeks’ gestation to assess the number of fetuses, fetal viability, presence of major fetal defects, and to take measurements of the crown–rump length (CRL) to date the pregnancy, and the nuchal translucency (NT). CRL and NT are combined with maternal age and the blood serum levels of free fraction of β-hCG and PAPP-A, measured at 9–12 weeks, to calculate the specific risk of the patient for the presence of trisomies 21, 18 and 13 [6]. If the risk is <1/270, the patient is informed that the risk for said aneuploidies is low and will be assessed at the second-trimester morphologic scan at 19–21 weeks. If the risk is between 1/50 and 1/270, it is considered an intermediate risk for aneuploidies, and the patient is offered the choice between an IT or cfDNA testing. In the case of an FCT ≥ 1/50, or if there are major fetal defects or an NT ≥ 3.5 mm, the patient is informed that, apart from a high risk for aneuploidies, there are also higher risks for other chromosomic and sub-chromosomic anomalies, and an IT with micro-array analysis is recommended. If the patient does not consent to an IT, cfDNA is offered.

Patients who chose to perform a cfDNA test gave their written informed consent prior to blood sampling (20 mL), which was collected in Roche Cell-Free DNA Collection Tubes (Roche, Pleasanton, CA, USA). Tubes were sent without processing to the cfDNA laboratory in Madrid, Spain. A directed analysis of cfDNA for the detection of trisomies with the prenatal Harmony^®^ test (Roche Diagnostics, Basel, Switzerland) was performed. A risk of ≥1% is considered as high risk, and patients with said result are recommended to consider IT for prenatal genetic diagnosis confirmation. Patients whose result is low-risk are informed of the low risk for trisomies and will be assessed at the 19–21-week morphologic scan. Patients whose testing does not offer results are offered a second blood sampling, and in the case of a repeated inconclusive result, are offered IT.

### 2.2. Screening Performance

Detection rates (DRs) and false positive rates (FPRs) and their 95% confidence intervals (CIs) were calculated for the current strategy, with a high-risk cut-off point of 1/50. DRs and FPRs, with their 95 CI%, were also calculated for the proposed strategy with a high-risk cut-off point of 1/10.

### 2.3. Postnatal Results

The postnatal result was evaluated based on the prenatal genetic karyotype analysis or the postnatal evaluation of the newborn. Postnatal cases with suspicion of chromosomal anomalies were followed up until 6 months after the delivery.

### 2.4. Statistical Analysis

A sample size of 109 pregnant women with chromosomal disorders (sample size determined by the nQuery Advisor 4.0 program) was needed to achieve an expected sensitivity of 90% (vs. 80% set for the chromosomal disorder) for a population of 25,200 pregnant women, with an α-error of 0.05, a prevalence of chromosomal disorders of 0.16% (1 in 600), and a power of 80% in the bilateral test. The descriptive data were described as frequencies and percentages. The IBM SPSS statistics software version 26 (IBM, Armonk, NY, USA) was used for this purpose.

## 3. Results

During the timeframe included in the study, the data of 25,920 patients were collected who attended their first-trimester hospital visit at one of the three participant hospitals. Out of these, 25,374 (97.9%) consented to the cfDNA contingent screening for aneuploidies, while 546 (2.1%) declined the screening. In the total sample, there were 201 (0.7%) cases of genetic anomalies, including 117 cases of trisomy 21 (0.5%).

### 3.1. Performance of the First-Trimester Contingent Screening with the Current Strategy

The results of the application of the current screening strategy with a high-risk cut-off point of 1/50 are displayed in Table 1. We detected 109 (93.2%; CI95%: 88.6–97.7) out of 117 cases of trisomy 21; 29 (87.9%; IC95%: 76.7–99.0) out of the 33 cases of trisomy 18 and 11 (84.6%; IC95%: 65.0–100) out of the 13 cases of trisomy 13. Regarding other genetic anomalies, we detected 27 (71.1%; IC95%: 56.6–85.5) out 38, and 307 (1.2%; IC95%: 1.1–1.3) ITs were performed. When including the results of the 20-week morphologic scan, the detection rate increased to 96.6% (113/117; IC95%: 93.3–99.9) for trisomy 21, 100% (33/33; IC95%: 100-100) for trisomy 18 and 100% (11/11; IC95%: 100-100) for trisomy 13, performing a total of 644 (2.5%; IC95%: 2.3–2.7) ITs. In 280 cases (1.1%), we detected major fetal defects, a TN thickness ≥ 3.5 mm or an FCT risk above 1/50, composing the high-risk group. In this group, there were 154 aneuploidies, including 95 cases of trisomy 21, 27 of trisomy 18 and 11 of trisomy 13. A total of 230 patients chose to undergo an IT, while 44 patients chose cfDNA testing. In the latter group, 10 had a high-risk result, and three had an inconclusive result after two samplings. Thus, the total number of ITs was 243.

In the second group, with an intermediate risk between 1/50 and 1/270, without major fetal defects or increased TN thickness, there were 1019 (4%) cases. Of these, 14 (1.4%) had a trisomy 21, two (0.2%) had a trisomy 18, and six (0.6%) had other anomalies.

In the low-risk group, with an FCT below 1/270, without fetal defects or increased NT thickness, there were 25 genetic anomalies, with eight, four and two cases of trisomies 21, 18 and 13, respectively, and 11 other anomalies. The number of ITs performed due to an FCT result was 307 (1.2%), out of which 243 (1.0%) belonged to the high-risk group and 64 (0.3%) to the intermediate-risk group. Adding the ITs performed due to existing defects in the 20-week’ morphologic scan, the total number of ITs was 644 (2.5%).

A total of 1092 cfDNA tests were performed: 46 (4.2%) in the high-risk group, 992 (90.8%) in the intermediate-risk group, and 54 (4.9%) in the low-risk group. Out of all of them, five patients had an inconclusive result after the first sampling and underwent an IT. Out of the six patients who had an inconclusive result after the second sampling, four underwent an IT.

### 3.2. Theorical Performance of the First-Trimester Contingent Screening with the Proposed Strategy

The proposed strategy for the first-trimester contingent screening would lower the cut-off point for the high-risk group from 1/50 to 1/10. Thus, patients with an FCT between 1/10 and 1/50, without major fetal defects or an increased NT thickness, would pass over from the high-risk to the intermediate-risk group. First, we assessed the patients that composed this group, as can be seen in Table 2. There were 83 patients, amongst whom 19 (22.9%) had a trisomy 21. There were not any cases of trisomies 18 and 13, and only three (3.6%) cases with other genetic anomalies. In this group, we performed 59 invasive techniques and 32 cfDNA tests.

In Table 3, we present the theorical results that we would have obtained if the proposed strategy had been implemented. With said strategy, there would have been 19 cases of trisomy 21 that would have passed over from the high-risk group to the intermediate-risk group. Knowing that the DR of cfDNA testing for trisomy 21 is 99.7%, we might assume that all 19 cases of trisomy 21 would have been detected with said test. Thus, the DR for trisomy 21 would have stayed at 93.2%, increasing to 96.6% after the 20-week morphologic scan. Assuming that all 61 eukaryotic cases, as well as the three cases with other genetic anomalies, had obtained a low-risk result in the cfDNA testing, only 19 ITs would have been performed in the trisomy 21 cases with a high-risk result. Therefore, a total of 267 (1.1%) ITs would have been performed with the proposed strategy, while the current strategy had a total of 307 (1.2%) ITs. In addition, the false positive rate would have stayed at 5.2% in both scenarios.

## 4. Discussion

The emergence of cfDNA testing has entailed a major advance in the field of prenatal diagnostics of aneuploidies, given its high DR for trisomies without the risks associated with invasive techniques. However, its expensive costs have put a toll on its implementation as a universal screening. Several authors have evaluated various contingent screening strategies using different cut-off points for the high-risk group [11,12,13,14,15].

Currently, the SEGO recommendations are based on a strategy using a 1/50 cut-off point for the high-risk group. Gil MM et al. [7,8] report a DR and FPR of cfDNA testing that are 99.7% (95% CI, 99.1–99.9%) and 0.04% (95% CI, 0.02–0.07%), respectively. This is the reason why international scientific societies such as the SEGO propose its use in the case of the high-risk group. Nonetheless, we know that it is below the 1/10 risk where a higher percentage of aneuploidies exist, as published by Sánchez-Duran et al. [13]. In order to decrease the rate of ITs, our study has evaluated what would have happened if we had lowered the cut-off point for the high-risk group from 1/50 to 1/10. 

Our results show that the group with an FCT between 1/10 and 1/50 only had 19 cases of trisomy 21 and only three other different anomalies. If these patients had been offered a cfDNA test in the first place, just like the intermediate-risk patients, we can assume that all trisomy 21 cases would have been detected, given the 99.7% DR of cfDNA. This way, the DR for trisomy 21 would have stayed at 93.2%, the same as the current strategy. In addition, the other 40 patients would not have undergone an IT, and the total rate of ITs would have decreased from 1.2% to 1.1%. While these changes might seem small, it would have been an improvement in patient care, as these 40 patients would have avoided the associated risk of an invasive procedure. This circumstance is in consonance with the findings of the study developed by Persico et al. [16], in which it is suggested that a policy of selecting a subgroup for invasive testing and another for cfDNA testing would substantially reduce the invasive procedures and retain the ability to diagnose most of the observed aneuploidies.

One of the main strengths of our study is the large sample size, as well as the promising results. Nonetheless, it also has its limitations, with the retrospective design being one of them. A future prospective study to evaluate the real implications of the application of this strategy shall confirm these results, as well as evaluate the economic implications.

## 5. Conclusions

The contingent screening of aneuploidies based in the result of the FCT, offering the analysis of cfDNA to patients with an intermediate risk after lowering the cut-off point from 1/50 to 1/10, is a valid alternative, which might keep the current detection rates and avoid the complications associated with invasive testing.

## Figures and Tables

**Table 1 biomedicines-10-02579-t001:** Performance of the first-trimester contingent screening with the current strategy.

	High-Risk Group				Intermediate-Risk Group				Low-Risk Group			
	(FCT ≤ 1/50, NT ≥ 3.5 mm, Fetal Abnormality)				(FCT 1/50–1/270)				(FCT ˂ 1/270)			
	T21	T18	T13	Others	T21	T18	T13	Others	T21	T18	T13	Others
N	95	27	11	21	14	2	0	6	8	4	2	11
Description				5x Monosomy X; 1x peripheric Cr8 inversion; 1x Trisomy 7; 5x Triploidy; 1x microdeletion Cr16; 3x Noonan Sd; 1x Klinefelter Sd; 2x Trisomy 16; 1x TAR Sd (Del 1q21.1); 1x DiGeorge Sd				2x Monosomy X; 1x Mosaic monosomy X; 1x t(1;13); 1x paracentric Inv(7); 1x Prader Willi Sd				2x Monosomy X; 1x Trisomy 12; 1x Triploidy; 1x Inv(9); 1x t(9;15); 1x Mosaic trisomy 16; 1x DiGeorge Sd; 1x ROB t(13;14); 1x Del(17); 1x Trisomy X
Identified by FCT ≥ 1/50	65	14	5	9 (2x Monosomy X; 4x Triploidy; 1x Peripheric inversion Cr8; 1x Trisomy 7; 1x Klinefelter Sd)	-	-	-	-	-	-	-	-
Identified by NT ≥ 3.5 mm	17	4	3	5 (2x Monosomy X; 3x Noonan Sd)	-	-	-	-	-	-	-	-
Identified by major fetal defects	13	9	3	7 (1x Monosomy X; 1x Triploidy; 1x microdeletion Cr16; 2x Trisomy 16; 1x TAR Sd (Del 1q21.1); 1x DiGeorge Sd)	-	-	-	-	-	-	-	-
Termination of pregnancy	146				14				57			
Invasive testing	243				64				337			
cfDNA	46 (10 High-risk; 3 inconclusive)				992 (13 High-risk; 8 inconclusive)				54 (54 Low-risk)			

FCT: first-trimester combined test; NT: nuchal translucency; T21: Trisomy 21; T18: Trisomy 18; T13: Trisomy 13; Sd: Syndrome; cfDNA: cell-free DNA; TAR: Thrombocytopenia-absent radius.

**Table 2 biomedicines-10-02579-t002:** Group with an FCT between 1/10 and 1/50.

	FCT 1/10–1/50			
	T21	T18	T13	Others
N	19	0	0	3
Description				1x Trisomy 16; 1x TAR Sd (Del 1q21.1); 1x DiGeorge Sd
Termination of pregnancy	21			
Invasive testing	59			
cfDNA	32 (5 High-risk; 1 inconclusive)			

cfDNA: cell-free DNA.

**Table 3 biomedicines-10-02579-t003:** Comparative performance of both strategies.

	Current Strategy (High-Risk Cut-Off Point 1/50)	Proposed Strategy (High-Risk Cut-Off Point 1/10
High-risk group (N)	280	197
Trisomy 21	95	76
Trisomy 18	27	27
Trisomy 13	11	11
Intermediate-risk group (N)	1019	1102
Trisomy 21	14	33
Trisomy 18	2	2
Trisomy 13	0	0
FCT Detection rates (%, CI 95%)		
Trisomy 21	93.2% (88.6–97.7)	93.2% (88.6–97.7)
Trisomy 18	87.9% (76.7–99.0)	87.9% (76.7–99.0)
Trisomy 13	84.6% (65.0–100)	84.6% (65.0–100)
Detection rates after the 20-week scan (%, CI 95%)		
Trisomy 21	96.6% (93.3–99.9)	96.6%
Trisomy 18	100.0% (100-100)	100.0%
Trisomy 13	100.0% (100-100)	100.0%
False positive rate	5.2%	5.2%
Invasive testing	307	267

FCT: first-trimester combined test.

## Data Availability

Not applicable.

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
