# Peer review of "Does a High-Risk (>1/50) Result for First-Trimester Combined Screening Always Entail Invasive Testing? Which Patients from This Group Might Benefit from cfDNA Testing?"

_biomedicines, 2022, doi:10.3390/biomedicines10102579_

Round 1

Reviewer 1 Report

The manuscript by Garcia-Jimenez et al., is an interesting study aiming to investigate potential criteria for an alternate approach with cfDNA testing in pregnant women to evaluate the theorical performance of lowering the cut-off point for 19 the high-risk group from 1/50 to 1/10. This is an interesting study with potential clinical application in the fields of diagnostics and optimized healthcare intervention.

The manuscript is well-written and flows easily. Minor issues with typos and optimized syntax exist and proofreading potentially by an English native speaker is suggested.

The reviewer would like to offer the following points for consideration by the authors:

1. What were the inclusion and exclusion criteria for study participation?

2. How was the number of patients/participants determined (eg power calculation etc).

3. Along those lines to what extent and how confidently are the results/findings and conclusions translatable to the general public? It appears that moving from the 1/50 to the 1/10 risk cut off does not lose many true cases but how strong are the data to propose that approach to millions of pregnancies?

4. To what extent are the results of the cfDNA validated against an invasive method?

Reviewer 2 Report

Manuscript 'Does a high-risk (>1/50) result for first-trimester combined screening always entail invasive testing? Which patients from this group might benefit from cfDNA testing?' is a well written paper with the aim to propose a different model for prenatal testing. By lowering cut off point, authors proved that it is possible to lower rate of invasive testing, but still to keep the current detection rates.

Although it is retrospective study, it could help to tailor the future screening strategies, therefore I find it worth publishing and interesting to the readers. I recommend it to be published in a present form.

Round 2

Reviewer 1 Report

The authors made a reasonable effort in addressing reviewer's comments.